# DUA-MQTT: A Distributed High-Availability Message Communication Model for the Industrial Internet of Things

**DOI:** 10.3390/s25165071

**Published:** 2025-08-15

**Authors:** Anying Chai, Wanda Yin, Mengjia Lian, Yunpeng Sun, Chenyang Guo, Lei Wang, Zhaobo Fang

**Affiliations:** 1College of Information Science and Engineering, Shenyang University of Technology, Shenyang 110870, China; chaianying@sut.edu.cn (A.C.); 18841957080@163.com (W.Y.); chenyang_guo1999@163.com (C.G.); 15727597129@163.com (L.W.); fangzhaobo2023@gmail.com (Z.F.); 2School of Mathematics and Information Engineering, Longyan University, No.1 Dongxiao North Road, Xinluo District, Longyan 364012, China; 3School of Electrical Engineering, Shenyang Institute of Engineering, No.18 Puchang Road, Shenbei New District, Shenyang 110136, China; sunyp@sie.edu.cn

**Keywords:** Industrial Internet of Things, OPC UA, distributed systems, information modeling, MQTT

## Abstract

With the rapid development of the Industrial Internet of Things (IIoT), the scale of industrial equipment has expanded, leading to an increasing diversity of communication protocols and a significant rise in data transmission volume within industrial networks. Traditional communication systems, constrained by concurrency and throughput limitations, struggle to meet the demands of massive data transmission. To address this issue, this paper proposes a distributed high-availability message communication model for IIoT (DUA-MQTT) based on the OPC UA architecture. It integrates the distributed MQTT protocol to enhance concurrency and throughput performance. Additionally, to improve the information processing capability of the proposed model, this paper designs an information-modeling model based on industrial unstructured text data (MAC-GC), which generates structured data nodes that comply with the OPC UA information model specification through hierarchical annotation, accurately mapping device functions and attributes. Experimental results show that, compared with traditional communication models, the DUA-MQTT model reduces end-to-end latency by 28.6% and increases throughput by 22.2%, effectively enhancing the concurrency of data transmission. In terms of information-modeling capabilities, MAC-GC outperforms other models in accuracy (0.9701), recall (0.9601), and F1 score (0.9651), effectively improving the utilization efficiency and modeling accuracy of unstructured data.

## 1. Introduction

With the in-depth development of the new round of technological revolution and industrial transformation, emerging information and communication technologies, such as the Internet of Things (IoT), cloud computing, and artificial intelligence, are accelerating their integration, continuously driving the manufacturing industry towards higher levels of automation and intelligence [1]. Among them, the Industrial Internet of Things (IIoT) accelerates the deep integration of industry and information technology. It significantly promotes transforming and upgrading traditional manufacturing systems towards digitalization, networking, and intelligence [2,3]. IIoT connects sensors, devices, and industrial systems, enabling real-time data collection, status monitoring, and intelligent scheduling in production, effectively enhancing the transparency and response efficiency of the manufacturing process and providing key technical support for sustainable development in the industrial sector. However, as the scale of IoT devices continues to expand, existing industrial communication systems gradually reveal performance bottlenecks and inefficiencies when facing critical scenarios, such as large-scale data concurrency, low latency transmission, and high throughput requirements [4,5,6]. Firstly, at the communication architecture level, the traditional client/server model makes it challenging to meet the demands of large-scale distributed device access and high-concurrency data transmission. Secondly, at the network performance level, achieving stable communication with low latency and high throughput in dynamic and complex industrial environments remains a core challenge that needs to be addressed urgently. Additionally, at the data-processing level, the large volume of unstructured text data from field devices has increased the complexity and uncertainty of the information-modeling process.

In data interactions among multi-source heterogeneous devices, traditional communication protocols have obvious limitations in flexibility and interoperability, and it is challenging to meet the real-time requirements of industrial scenarios [7]. As a widely used industrial communication standard, OPC UA performs well in device interconnection and data exchange, especially in cross-platform compatibility and unified semantic modeling support [8,9]. The performance of the OPC UA system may degrade under high concurrency scenarios. In certain load conditions, the average transmission latency at QoS Level 2 can increase to several seconds. The single-agent configuration of MQTT, while lightweight and efficient, can become overloaded and introduce a single point of failure in large-scale deployments, limiting its suitability for mission-critical applications unless enhanced with redundancy mechanisms [10]. To compensate for these deficiencies, the integration of OPC UA and MQTT has gradually become an effective optimization solution. While not intended to replace existing solutions such as OPC UA PubSub or AMQP, the DUA-MQTT model integrates the semantic modeling strengths of OPC UA with the lightweight message delivery of MQTT, offering a practical alternative for high-concurrency and resource-constrained industrial scenarios. OPC UA provides standardized semantics and cross-platform interoperability, while MQTT ensures low-latency and low-bandwidth message distribution. This synergy offers improved scalability and throughput, particularly in scenarios involving large-scale heterogeneous devices and frequent message bursts, where lightweight transport mechanisms and distributed brokers provide performance advantages. OPC UA has strong information-modeling capabilities, providing detailed semantic descriptions and standardized data interaction methods. In contrast, MQTT, as a lightweight publish/subscribe protocol, has the advantages of low bandwidth occupation, high transmission efficiency, and good scalability, making it suitable for high-load industrial network environments. By combining the interconnection and intercommunication capabilities of OPC UA with the efficient data distribution mechanism of MQTT, it is possible to achieve a more efficient data transmission performance while ensuring data interoperability, which is suitable for low-bandwidth and high-delay network environments [11,12].

The current mainstream OPC UA-MQTT integration architecture typically adopts a centralized proxy model, relying on a single server for unified data routing and distribution. However, with the rapid increase in the number of industrial devices and the continuous rise in communication traffic, the centralized proxy is prone to become a system bottleneck, not only weakening the network’s scalability but also increasing the risk of single-point failure, which seriously affects the system’s fault tolerance and operational stability [13]. In industrial network environments, in addition to structured numerical data collected by sensors and PLCs, a large amount of unstructured text data describing the functions, states, and attributes of devices also has significant analytical value. However, the existing OPC UA-MQTT architecture mainly focuses on modeling and managing structured data. Although it can uniformly schedule information from sensors and control systems and provide a standardized modeling mechanism [14], it still has insufficient modeling capabilities and low processing efficiency when dealing with large-scale unstructured industrial data. Moreover, industrial production systems usually comprise multiple heterogeneous devices involving different communication protocols and data formats. Achieving high-concurrency, low-latency data transmission in multi-source heterogeneous industrial networks while ensuring high throughput performance and reducing the complexity of information modeling has become a key challenge that needs to be urgently addressed in the distributed communication system of the industrial Internet of Things [15].

In response to the above issues, this paper proposes a distributed high-availability message communication model for the industrial Internet of Things, DUA-MQTT. It builds an architecture for device interconnection and interoperability. It also introduces the distributed MQTT protocol to enhance the system’s concurrent processing capacity and data transmission efficiency. Additionally, to meet the information-modeling requirements of unstructured data in industrial communication systems, this paper designs an information-modeling model for industrial unstructured text data, MAC-GC. This model uses hierarchical annotation technology to generate structured data nodes that comply with the OPC UA standard, and mapping the attribute information of devices achieves efficient information modeling. Experimental results show that DUA-MQTT demonstrates superior performance in high-load and large-scale application environments, meeting the requirements of high concurrency and reliability in the industrial Internet of Things. Meanwhile, MAC-GC improves the processing efficiency and modeling accuracy of unstructured data, complementing the traditional OPC UA architecture, which is primarily optimized for structured information.

The main contributions of this paper are as follows:(1)Based on the traditional client/server communication architecture, this paper proposes the DUA-MQTT distributed communication model, which introduces a multi-agent collaboration mechanism and the distributed MQTT protocol, to achieve efficient communication among heterogeneous devices. This model adopts a multi-agent mechanism to optimize resource utilization, combines load-balancing strategies to enhance system scalability and fault tolerance, effectively reduces communication latency, and improves transmission efficiency in high-concurrency environments.(2)To address the challenge of modeling unstructured text data in industrial scenarios, this paper proposes the MAC-GC modeling model, which is used to achieve high-quality semantic expression, context feature capture, and label sequence decoding. This model can accurately extract key entities and attribute information from industrial texts and construct information model nodes that conform to the OPC UA standard, significantly enhancing the expression ability and modeling accuracy of unstructured data.(3)The DUA-MQTT model has been experimentally verified to possess a superior concurrent transmission performance in high-load, large-scale industrial Internet of Things environments. The distributed architecture significantly enhances system throughput and reduces end-to-end communication latency. Meanwhile, the MAC-GC model demonstrates higher processing efficiency and semantic understanding capabilities in information-modeling tasks involving unstructured data.

The remaining part of this article is organized as follows: Section 2 reviews the research progress of OPC UA and MQTT protocols, analyzes the key issues of protocol integration, and discusses the current status and challenges of unstructured data processing and information modeling. Section 3 introduces the architecture of the DUA-MQTT model, including core modules such as data analysis, unstructured data processing, OPC UA address space management, and protocol integration, and proposes a distributed multi-agent communication strategy based on MQTT. Section 4 presents the experimental design and evaluation, verifying the high concurrent communication performance of DUA-MQTT and the modeling capability of MAC-GC in industrial unstructured data processing, and conducts a comparative analysis with existing methods. Section 5 summarizes the research results; discusses its application value in IIoT; and looks forward to future research directions in protocol optimization, data processing, and intelligent modeling.

## 2. Related Work

With the rapid development of IIoT, network access, and protocol integration technology in smart factories has become an important research direction in the industrial field. Xia et al. [16] analyzed smart factories’ heterogeneity of devices, communication methods, and access networks. They proposed a heterogeneous network integration architecture based on the gateway and Software-Defined Network (SDN). This architecture deeply discusses the challenges of network management, equipment upgrading, and data fusion and provides theoretical support for optimizing network architecture in smart factories. However, with the introduction of various industrial communication protocols, the compatibility and efficient integration between different protocols are still the key issues in realizing intelligent manufacturing.

As an industrial communication standard, OPC UA is prominent in device interconnection due to its cross-platform data exchangeability and unified semantic modeling function. However, in some high-concurrency and large-throughput scenarios, traditional OPC UA systems based on centralized architectures may encounter scalability and responsiveness limitations. To solve this problem, the lightweight publish/subscribe protocol MQTT was introduced into the industrial scene. MQTT has emerged as an effective complementary protocol in real-time data transmission, especially in networks with low bandwidth and high latency, due to its lightweight publish/subscribe architecture [17].

Several studies have proposed optimization schemes to combine MQTT protocol with OPC UA. These studies show that integrating MQTT and OPC UA can significantly improve the concurrency and throughput of data transmission and effectively alleviate the performance bottleneck caused by the traditional centralized broker architecture. Liu et al. [18] simplified the parameter setting of OPC UA and MQTT and improved communication efficiency by introducing a graphical configuration interface. Nast et al. [19] proposed to integrate the MQTT-SN protocol into OPC UA and used its lightweight characteristics to improve the high-frequency and low-latency message transmission ability of OPC UA in the industrial Internet of Things, providing a scalable solution for real-time data management in industrial automation systems. The BORDER benchmark framework proposed by Longo et al. [20] evaluates the performance of MQTT in distributed agents through simulators and Docker containers, which provides an essential reference for protocol optimization. In addition, the MQTT-A protocol proposed by Buccafurri et al. [21] introduces a peer-to-peer (P2P) cooperation mechanism to improve the anonymity and security of data transmission, which is especially suitable for industrial scenarios with high data privacy requirements. Shi et al. [22] discussed the industrial Internet of Things based on MQTT and OPC UA protocols, which realizes data acquisition, control, and information interaction in the industrial production process through intelligent terminal devices, edge-computing algorithms, and cloud service platforms, to improve resource utilization and promote the development of smart manufacturing.

In addition to optimizing communication mechanisms, security remains a fundamental concern for any IIoT system. Sheng et al. [23] conducted a comprehensive survey on IIoT device identification through network traffic fingerprinting, categorizing methods into packet-level, flow-level, and business-level fingerprinting. They also highlighted challenges such as limited generalization, obscure feature spaces, and scarce datasets, all of which pose potential threats in heterogeneous IIoT environments. Feng et al. [24] reviewed firmware vulnerability analysis methods, classifying approaches based on firmware unpacking, static and dynamic analysis, and emulation-based strategies. Their study revealed the difficulties in building trusted firmware analysis pipelines due to device heterogeneity and the use of closed-source code. These insights provide valuable references for enhancing device trust verification and network traffic integrity monitoring in the design of secure IIoT communication frameworks such as DUA-MQTT.

Regarding communication optimization of industrial equipment, Bauer et al. [25] proposed an OPC UA hardware implementation scheme based on FPGA, which proved the high efficiency of FPGA hardware in industrial automation. The multi-agent data distribution scheme proposed by Cho et al. [26] reduces network delay. It improves the data transmission efficiency by considering node correlation and data priority, which provides an essential reference for resource optimization and real-time data processing in the industrial Internet of Things. Chai et al. [27] proposed a QoS prediction model based on EMD-mLSTM to improve the dynamic QoS prediction accuracy of IIoT, effectively reduce communication delay, and support brilliant factory service selection and optimization. Yang et al. [28] proposed a dynamic random multipath routing method (DRMRM) to optimize the energy efficiency and network lifetime of linear Wireless sensor networks (LWSNs). By combining node depth and residual energy model, DRMRM avoids the energy consumption dependent on the routing table.

In addition to communication protocol optimization, the processing of unstructured data has increasingly attracted attention in the industrial domain. Unstructured data—such as the functional and attribute information of devices—contain rich and valuable information. However, due to the lack of unified formatting standards and semantic definitions, OPC UA encounters significant challenges when directly handling such data [29]. Effectively extracting insights from these data to enhance industrial intelligence and support data-driven decision-making has thus become a crucial research focus. Furthermore, with the rapid proliferation of IoT devices, system security has emerged as a key consideration in communication model design. Gide et al. [30] proposed a real-time hybrid intrusion detection framework based on KNN and neural networks, which demonstrates strong performance in detecting DoS/DDoS attacks in MQTT-based environments.

At present, the OPC UA protocol has achieved remarkable results in device interconnection, data exchange, and information modeling, especially in cross-platform communication and information interoperability [31]. However, OPC UA mainly targets structured data and has limited processing ability for unstructured data, posing challenges for broader application in complex industrial environments [32]. Therefore, some studies proposed combining the emerging data-processing technology with OPC UA. Muniraj et al. [33] proposed an improved machine communication framework based on the OPC UA protocol, which optimized decision-making and command distribution through vertical and horizontal communication to improve the intelligence level of the industrial Internet of Things. To solve this problem, Wenbin Dai et al. [34] proposed an automatic information model generation method based on IEC 61499 and OPC UA, which converts the design-time model into the run-time model through model transformation rules to support the integration of edge devices and industrial cloud platforms. Axel Busboom et al. [35] proposed a method to simplify the construction of information models, reduce the dependence on in-depth domain knowledge, and improve development efficiency. Martins et al. [36] explored the concept of Industry 4.0 from the die-and-mold industry perspective. They proposed an OPC UA-based processing equipment-monitoring solution, which enables standardized high-level communication by integrating different device interfaces into one system, providing comprehensive information for production managers to optimize the manufacturing process.

In this context, the research of combining natural language-processing technology and OPC UA protocol has gradually become a hot topic, aiming to improve the processing ability of unstructured data and the intelligence level of information modeling. By combining natural language-processing technology, such as BERT, with OPC UA protocol, the processing ability of unstructured data can be significantly improved, and the intelligence level of information modeling can be enhanced. This integration not only makes up for the shortcomings of traditional OPC UA in unstructured data processing but also promotes the development of intelligent manufacturing and industrial Internet of things and improves the intelligence and adaptability of the system. Similar challenges and solutions have received much attention from researchers in the biomedical field. Wang et al. [37] summarized the progress of pre-trained language models (PLMs) in the biomedical domain, proposed a taxonomy of PLMs, and explored their application to biomedical tasks, particularly on text, electronic health records, protein and DNA sequences, etc. Similarly, BioBERT [38] has improved the performance of conventional NLP models in biomedical text mining by pre-training on large-scale biomedical literature, aiming at the challenges brought by domain-specific words and terms. However, its generalization ability and stability still need to be further improved. The MacBERT model proposed by Cui et al. [39] improves the performance of Chinese pre-trained models in multi-task learning by introducing the full-word mask strategy, which provides a theoretical basis and practical guidance for the research and application of Chinese natural language processing. Although these research results in the biomedical field provide reference for similar industrial scenario problems, they still need to be customized and adjusted for the unique needs of the industrial field. Especially in the industrial Internet of Things system, the processing and transformation of unstructured data into standardized structured information is the basis for intelligent decision-making and automatic management. The combination of BERT and OPC UA protocol provides a promising solution, which can effectively improve the intelligent data processing and information modeling capabilities and promote the development of industrial systems in a more efficient and thoughtful direction.

Aiming at the problems of low concurrency efficiency and poor throughput in processing massive data transmission in traditional industrial communication systems, this paper proposes a distributed high-availability message communication model for industrial Internet of Things (DUA-MQTT). The model uses OPC UA as the underlying architecture and combines distributed MQTT protocol to improve the concurrent processing ability and data transmission efficiency. In addition, according to the characteristics of information modeling in the communication model, we designed an information-modeling model (MAC-GC) suitable for industrial unstructured text data. Through the hierarchical annotation technology, the model generates structured data nodes conforming to the OPC UA specification and comprehensively maps the device function and attribute information to realize efficient information modeling. DUA-MQTT demonstrates notable performance improvements in high-load and large-scale application scenarios, offering enhanced concurrency and reliability in comparison to traditional single-agent architectures. At the same time, the MAC-GC model we designed overcomes the limitations of the traditional OPC UA architecture by improving the utilization efficiency and modeling accuracy of unstructured data, and it provides technical support for information management and analysis in IIot.

## 3. Distributed and Highly Available Messaging and Communication Model for Industrial IoT

### 3.1. Overall Architecture

In the IIoT field environment, where large-scale device deployment and massive heterogeneous data transmission are prevalent, the system often faces processing efficiency and throughput challenges. This paper proposes a distributed high-availability message communication model for industrial IoT, as illustrated in Figure 1. The model comprises four main modules: data analysis, unstructured data processing, OPC UA address space, and protocol integration and message routing.

The data-processing module receives and initially processes data from various sources. It initially categorizes the data into structured or unstructured types, wherein structured data, such as sensor measurements (e.g., temperature, pressure), is forwarded to the information model definition module for further refinement and subsequently integrated into the OPC UA address space as structured data nodes. Unstructured data, comprising device functionality and attribute information, is then transmitted to the unstructured data-processing module for further analysis.

In the unstructured data-processing module, our designed MAC-GC model is introduced to identify entities and attributes within unstructured data. The model employs hierarchical annotation techniques to convert raw data into structured nodes that conform to the OPC UA information model specification. By leveraging its robust semantic modeling capabilities and precise sequence labeling mechanism, the MAC-GC model enhances the accuracy of processing unstructured data.

The OPC UA Address Space module stores processed data within the OPC UA address space while ensuring compliance with the OPC UA Information Model specification. This module organizes data structurally by creating standardized nodes, such as object and variable instances, to represent physical or logical entities and their associated attributes. Additionally, it supports dynamic updating and expansion of nodes, enabling real-time configuration and modification based on specific application requirements. By maintaining consistency and performance, the module ensures that nodes can rapidly reflect and update relevant data in response to equipment or system status changes.

The protocol integration and message-routing module comprises three sub-modules: protocol conversion, message routing, and publish/subscribe. The protocol conversion module converts complex data structures from the OPC UA address space module into a simplified format suitable for the MQTT protocol. Due to the differences in data representation between OPC UA and MQTT, the protocol conversion module facilitates adaptation between the two protocols. Reading node data in the OPC UA address space extracts information about objects, variables, and attributes. It converts it into a format compatible with MQTT protocol transmission, enabling data exchange between OPC UA and MQTT. The message routing module ensures that data is transmitted through the optimal path while avoiding network bottlenecks and improving data transmission efficiency. This module dynamically selects the best routing path by monitoring network topology, delay, and bandwidth in real time to forward data from the root node to subscribers efficiently. Additionally, it automatically adjusts the routing path according to network conditions to ensure reliable message transmission in various environments, thereby maintaining system stability and efficiency. The publish/subscribe module implements message publication and subscription functionality. It provides accurate matching between MQTT topics and subscribers by maintaining subscription information, enabling efficient message publication to the target topic. According to the QoS requirements of the MQTT protocol, this module also ensures reliable message transmission under different network conditions, thereby ensuring smooth message flow.

To further illustrate the integration mechanism between the OPC UA and MQTT components in this architecture, Figure 2 presents the internal structure and layered functions of the DUA-MQTT framework. Figure 2 depicts the architectural design of the DUA-MQTT model, clarifying the functional role of OPC UA within the hybrid framework and specifying how the model builds upon core elements of the OPC UA architecture while integrating distributed MQTT-based communication to form a cohesive and complementary system.

At its core, the OPC UA layer retains three foundational capabilities of the standard OPC UA architecture. These include the information-modeling mechanism, which supports the definition of industrial data types, hierarchical object structures, and semantic relationships; the node management functionality, which maintains the organization and lifecycle of nodes within the OPC UA address space; and the security services, which offer robust authentication, authorization, and encryption based on OPC UA’s security infrastructure.

The connection between OPC UA and MQTT is established through a protocol bridge layer, which facilitates seamless interaction between the two domains. This layer is responsible for transforming OPC UA node structures into MQTT-compatible message formats while preserving their semantic context. It also manages intelligent message-routing strategies, ensuring efficient path selection and load balancing across the network. Additionally, it provides a mapping between OPC UA services and MQTT’s Quality of Service (QoS) levels to support reliable publish/subscribe communication based on topic hierarchies.

In terms of data handling, the architecture adopts a dual-path processing strategy for structured and unstructured data. Structured data from field devices is transmitted directly to the OPC UA layer via the data analysis module, where it is semantically modeled and encoded. In contrast, unstructured industrial text—such as logs, descriptions, or maintenance reports—is first processed by the MAC-GC model to extract structured semantic content, which is then integrated into the OPC UA information model. This ensures that all types of industrial information benefit from the standardized representation capabilities of OPC UA before being disseminated across the MQTT-based distributed broker network.

Overall, this architecture shows that DUA-MQTT employs OPC UA mainly for semantic representation and information modeling while relying on MQTT for data transport, message dissemination, and distributed coordination. This division of roles results in a hybrid system that combines the semantic expressiveness and industrial compatibility of OPC UA with the lightweight, high-throughput communication capabilities of MQTT.

### 3.2. Distributed Multi-Agent Collaborative Communication Strategy Based on MQTT

Due to its centralized architecture, the traditional MQTT protocol has exposed bottlenecks of low concurrent efficiency and insufficient throughput when dealing with large-scale devices and data streams. This paper introduces a distributed MQTT communication strategy based on a multi-agent collaboration mechanism to overcome these limitations. This strategy deploys multiple collaborative agent nodes and adopts a distributed agent architecture, effectively avoiding the single point of failure risk of traditional centralized systems and enhancing the system’s fault tolerance and scalability. To initialize the network, each agent periodically exchanges control information—including CPU performance, memory size, IP address, and message delay metrics—via MQTT PINGREQ/PINGRESP messages. This information is used to evaluate agent capability and determine the optimal root agent through a deterministic selection process. During topology construction, agents use a shared loop detection mechanism based on routing state tables to avoid redundant links and collaboratively form a loop-free acyclic structure. Real-time synchronization is maintained using MQTT’s KEEPALIVE mechanism, ensuring that topology updates and capability changes are propagated promptly across all agents. The distributed network structure is shown in Figure 3.

In the agent network construction phase, multi-agent network construction is introduced, and the original mechanism of MQTT is used to expand and optimize effectively, which improves the protocol’s stability and data transmission efficiency. The communication process diagram is shown in Figure 4.

Firstly, the key control information, including the root agent’s IP address, the agent capability value *Z*, and the path cost P, is transferred between the agents through the PINGREQ message of MQTT. The agent capability value *Z* is formulated as follows (1):(1)$$\begin{array}{c} Z=\alpha S+\beta R \end{array}$$
where *S* is the agent’s CPU speed, *R* is the memory size, and α and β are weight parameters used to regulate the influence of both on the ability value. The root agent is selected using an optimization algorithm based on resource capacity, considering hardware performance indicators such as CPU speed and memory size. When multiple agents have the same capability value, the agent with the lowest IP address is selected as the root agent to ensure the fairness and certainty of the selection process.

In the connection and signaling phase, the agents establish the initial communication connection by sending MQTT CONNECT messages with specific protocol version bytes and use the KEEPALIVE mechanism of MQTT to exchange PINGREQ/PINGRESP messages regularly to update the network topology information dynamically. Through these signaling messages, the system can estimate the real-time round-trip delay (RTT) between agents and update the path cost P based on the RTT. The RTT formula is as follows (2):(2)$$\begin{array}{c} RTT=T_{\mathrm{response}}-T_{\mathrm{request}} \end{array}$$

Trequest is the time to send the request, and Tresponse is the time to receive the response. At the same time, the non-root agent selects the best forwarding path according to the path cost and resource capacity to construct the acyclic network.

In the operational phase of the broker, while maintaining the client support features of the traditional MQTT protocol, the message-passing performance is improved by an efficient message distribution mechanism. As the central hub of the network, the root agent is responsible for forwarding incoming messages to all non-blocking connected non-root agents and then redistributing them according to the preferred routing path based on dynamically optimized route selection criteria, such as stability, bandwidth utilization, and delay. The system supports quick activation of blocking connections as a fallback mechanism in case of failures, ensuring network reliability and stability while preventing message loops.

To prevent message duplication and routing loops, each message is assigned a globally unique Message Identifier (MID) upon generation. All agents maintain a lightweight in-memory Message-ID cache with time-based eviction to filter repeated messages during routing. Additionally, a hop-count field is embedded in the MQTT header extension to restrict message traversal across excessive agents, further avoiding circular propagation. Regarding topic-level conflicts, the system enforces a hierarchical namespace policy, wherein wildcard subscriptions (e.g., factory/#) are only registered by the root agent, and child agents are restricted to concrete topic bindings (e.g., factory/machine1/status). This hierarchical delegation minimizes overlap, prevents subscription shadowing, and ensures consistent topic delivery semantics across all agents.

In the message-passing mechanism, the broker supports three QoS levels of the MQTT protocol to meet diverse application requirements. For Quality-of-Service (QoS) Level 0 (at most once), messages are transmitted without acknowledgments or retries, making it suitable for scenarios with minimal reliability demands. At QoS Level 1 (at least once), messages are guaranteed to be delivered at least once, though duplicates may occur, aligning with applications that require some reliability while tolerating redundancy. The system operates under QoS Level 2 (only once) for high-reliability and consistency-critical tasks such as industrial IoT, ensuring accurate and non-repetitive message delivery through a strict four-step handshake process. Additionally, the proxy network supports Last Will and Testament (LWT) message forwarding. Upon unexpected client disconnection, the broker automatically generates LWT messages to other clients, incorporating critical status information and pending tasks, thereby enhancing system recovery capabilities and ensuring timely measures are taken to maintain business continuity.

To preserve the semantics of each QoS level across distributed agents, the system implements a QoS consistency coordination mechanism. Specifically, during QoS 2 communication, each agent along the transmission path performs a local four-step handshake verification and synchronizes the delivery state to its neighbor agent. These acknowledgments are cached in a distributed state ledger to prevent duplicate delivery even after failover. Furthermore, in the case of LWT messaging, the designated backup agent of each client maintains a mirrored session state, ensuring that even if the hosting agent fails, the LWT message can be promptly published by the backup without loss. This redundant design reinforces both message reliability and failure resilience, aligning with MQTT’s quality guarantees in complex industrial networks.

Through mechanisms such as resource capability exchange, path cost assessment, and fault response among multiple agents, collaborative decision-making and autonomous management are achieved, thereby establishing a distributed communication system with adaptive capabilities.

In terms of fault handling, the system implements an adaptive topology reconstruction mechanism. When an agent detects an abnormal disconnection or message timeout—identified via KEEPALIVE failure or PINGRESP timeout—it broadcasts a fault signal embedded in a special PINGREQ message with a topology-change flag. Upon receiving this signal, all active agents initiate a root-agent re-election procedure, reassessing agent capability and updating the routing structure. To avoid message duplication during this process, each message is assigned a unique identifier (Message-ID), and agents maintain a distributed deduplication cache with limited TTL (time-to-live). Additionally, message loops are avoided by maintaining acyclic routing tables, and topic-level conflicts are resolved by enforcing hierarchical topic namespaces and only allowing the root agent to initiate wildcard-based subscription propagation. This ensures consistency across redundant agents while maintaining high availability. See Algorithm A1 in Appendix A for details.

### 3.3. Information-Modeling Method Based on Industrial Unstructured Text Data

Traditional information models typically rely on structured data tables with predefined formats, making them efficient for well-structured datasets. However, in real-world industrial applications, much of the information exists in unstructured textual formats, such as operation logs, maintenance notes, and alarm records. These sources lack consistent formatting, explicit semantics, and standardized relationships, which present challenges for automated information modeling. To address this, the proposed MAC-GC model employs a hierarchical semantic extraction framework that leverages domain knowledge and rule-based annotations. The modeling process includes three stages: semantic representation, contextual understanding, and structural transformation. In the semantic representation stage, unstructured industrial text is tokenized and segmented using an industry-specific vocabulary. In the contextual modeling phase, semantic roles such as entity, attribute, and value are identified through labeling rules grounded in domain ontologies. These roles are then aggregated into structured semantic triplets (e.g., 〈Pump A, status, failure〉) and are further mapped to a hierarchical format that conforms to the OPC UA node architecture. This transformation ensures that the resulting structured data can be seamlessly integrated into the OPC UA address space, where each entity becomes a node, attributes are modeled as variables, and values are stored as real-time attributes. This process bridges the semantic gap between unstructured language descriptions and machine-interpretable industrial information models. To illustrate with concrete examples, MAC-GC effectively processes various types of industrial unstructured texts, including maintenance logs (e.g., “Centrifugal pump P-101 inspection completed at 14:30. Bearing temperature 78 °C (high), vibration 4.2 mm/s. Lubricant replaced, alignment adjusted.”), alarm records (e.g., “URGENT: Boiler B-201 pressure exceeded safety threshold (142 bar > 135 bar) at 08:45:22.”), and operator notes (e.g., “Mixer M-305 producing unusual noise during operation. Speed reduced from 250 RPM to 180 RPM.”). When processing these texts, MAC-GC identifies entities, attributes, and values to construct semantic triplets like <Pump P-101, bearing temperature, 78 °C>, <Boiler B-201, pressure, 142 bar>, and <Mixer M-305, speed, 180 RPM>, which are then transformed into OPC UA-compatible node structures. The model structure is illustrated in Figure 5.

#### 3.3.1. Encoding Layer

The encoding layer constitutes the core semantic processing component of MAC-GC, employing a 12-layer Transformer encoder to convert unstructured industrial text into high-quality contextual representations suitable for subsequent entity recognition and attribute extraction tasks. MAC-GC integrates two pre-training tasks, namely Whole Word Masking (WWM) and Sentence Order Prediction (SOP), to enhance the model’s semantic modeling ability and the depth of text understanding. In the WWM task, the model performs masking operations at the level of complete words, effectively preserving word-level semantic boundaries and context consistency, while the SOP task, by learning the logical arrangement order between sentences, strengthens the model’s ability to model semantic relationships at the discourse level.

Regarding model structure, MAC-GC is built with a 12-layer Transformer encoder, whose core consists of a multi-head self-attention mechanism and a feedforward neural network. The multi-head self-attention mechanism can dynamically capture the global dependency relationships among elements in the sequence, thereby alleviating the semantic attenuation problem faced by traditional models when dealing with long-distance dependencies and generating more contextually relevant semantic representations.

In the pre-training phase, MAC-GC leverages large-scale general corpora, demonstrating notable advantages in Chinese language processing and complex industrial text-modeling tasks. This optimized design provides high-quality semantic feature support for subsequent natural language processing tasks. In the self-attention mechanism, each element in the input sequence is transformed by a linear transformation to generate a Query vector (Query, Q), Key vector (Key, K), and Value vector (Value, V). The vector calculation formula in the attention mechanism is as follows (3):(3)$$\begin{array}{cc} Q & =XW_{Q},\;K=XW_{K},\;V=XW_{V} \end{array}$$

Here, *X* is the embedding representation of the input sequence, and $W_{Q}$, $W_{K}$, and $W_{V}$ are the learnable weight matrices. The model measures the relevance of each element to other elements in the input sequence by computing the dot product of *Q* and *K*. The formula of the self-attention mechanism is as follows (4):(4)$$\begin{array}{cc} \mathrm{Attention}(Q,K,V) & =\mathrm{softmax}\left(\frac{QK^{⊤}}{\sqrt{d_{k}}}\right)V \end{array}$$

The computed dot-product results are scaled, and Softmax is normalized to generate an attention distribution. This distribution performs a weighted sum over *V* to aggregate contextual information and develop the final semantic representation. Through this mechanism, the semantic dependencies in the sequence can be efficiently captured, and richer semantic vectors can be generated, which provides high-quality feature representation for text tasks in the industrial field. The MAC-GC encoding layer is shown in Figure 6.

#### 3.3.2. Interaction Layer

MAC-GC simultaneously models the context information of the sequence through the forward and backward networks, which can effectively capture the semantic features with a strong correlation in industrial texts. By introducing two core mechanisms of update gate and reset gate, the number of parameters is reduced, and the calculation efficiency is improved.

Firstly, the reset gate $r_{t}$ controls the degree of fusion between the current input and historical information. Its calculation process is shown in Formula (5):(5)$$\begin{array}{cc} r_{t} & =\sigma \left(W_{r}\cdot \left[h_{t-1},x_{t}\right]+b_{r}\right) \end{array}$$

Here, $W_{r}$ represents the weight matrix of the reset gate, $\left[h_{t-1},x_{t}\right]$ indicates the concatenation of the hidden state from the previous time step and the current input, $b_{r}$ is the bias term, and σ is the sigmoid activation function. Next, the update gate $z_{t}$ determines the weighting distribution of old and new information in the hidden state, and its calculation method is shown in Formula (6):(6)$$\begin{array}{c} z_{t}=\sigma \left(W_{z}\cdot \left[h_{t-1},x_{t}\right]+b_{z}\right) \end{array}$$

Here, $W_{z}$ is the weight matrix of the update gate, $b_{z}$ is the bias term of the update gate, and σ is the activation function used for nonlinear transformation.

Based on the function of the reset gate, the candidate hidden state ${\tilde{h}}_{t}$ is generated, which serves as the candidate representation after fusing the current input with historical information at each time step, as shown in Formula (7):(7)$$\begin{array}{c} {\tilde{h}}_{t}=\tanh\left(W_{h}\cdot \left[r_{t}\circ h_{t-1},x_{t}\right]+b_{h}\right) \end{array}$$

Here, tanh is the tanh activation function, and $W_{h}$ and $b_{h}$ are the weight matrix and bias term, respectively.

Ultimately, the hidden state $h_{t}$ is generated by combining the update gate $z_{t}$ and the candidate hidden state ${\tilde{h}}_{t}$, as shown in Formula (8):(8)$$\begin{array}{c} h_{t}=\left(1-z_{t}\right)\circ h_{t-1}+z_{t}\circ {\tilde{h}}_{t} \end{array}$$

This step determines the final hidden state at the current time step, where $\left(1-z_{t}\right)$ and $z_{t}$, respectively, represent the weights of the old and new information.

The input sequence is respectively passed to the forward GRU and backward GRU, and through the computations of these two, the forward hidden state $h_{t}^{\mathrm{forward}}$ and backward hidden state $h_{t}^{\mathrm{backward}}$ are, respectively, generated. The final output is the concatenation of these two hidden states, as shown in Formula (9):(9)$$\begin{array}{c} h_{t}^{\mathrm{final}}=\left[h_{t}^{\mathrm{forward}},h_{t}^{\mathrm{backward}}\right] \end{array}$$

The bidirectional structure of MAC-GC enables it to simultaneously capture the forward and backward dependency information of sequences, thereby extracting the text’s semantic features more profoundly. In the industrial field, knowledge texts usually contain strong contextual correlations. Dynamic modeling of the semantic features of long sequences can effectively provide strong semantic support for the named entity recognition task.

#### 3.3.3. Inference Layer

In the output of the interaction layer, each step’s prediction is made independently, which may lead to inconsistent label sequences in sequence labeling tasks. We introduce an inference layer after the interaction layer to address this issue. We ensure the logical consistency between each label by constructing a transition probability matrix between labels, learning the dependencies among labels, and performing global inference on the entire sequence. The formula of its objective function is as follows (10):(10)$$\begin{array}{c} P\left(Y|X\right)=\frac{\exp\left(\sum_{i=1}^{n}\left(A_{y_{i-1},y_{i}}+S_{i,y_{i}}\right)\right)}{\sum_{Y^{\prime }\in Y\left(X\right)}\exp\left(\sum_{i=1}^{n}\left(A_{y_{i-1},y_{i}^{\prime }}+S_{i,y_{i}^{\prime }}\right)\right)} \end{array}$$

The input sequence $X=\{x_{1},x_{2},\ldots ,x_{n}\}$ represents the input data of the model, and the predicted label sequence $Y=\{y_{1},y_{2},\ldots ,y_{n}\}$ is the output label of the model; the label transition matrix $A_{y_{i-1},y_{i}}$ represents the probability of transitioning from label $y_{i-1}$ to label $y_{i}$, and the score of the feature representation output by the interaction layer with label $y_{i}$ is denoted by $S_{i,y_{i}}$. The set of all possible label sequences is represented by $\hat{Y}\left(X\right)$. During the training process, the transfer relationship between labels and the feature distribution of each label are learned by optimizing the conditional probability; in the decoding stage, the inference layer performs global inference on the output of the interaction layer to generate the label sequence *Y* and ensures the logical consistency of each label in the sequence.

During the training phase, the inference layer learns to determine the underlying principles governing label transitions and the probability distributions of feature scores by systematically optimizing the aforementioned conditional probabilities. The corresponding loss function formula is as follows (11):(11)$$\begin{array}{c} L_{CRF}=-\log P\left(Y|X\right) \end{array}$$

This objective function ensures that the model can efficiently learn the optimal global distribution of label sequences. By integrating the interaction layer’s context feature extraction capability with the reasoning layer’s global inference ability, the model’s performance in sequence-labeling tasks has been enhanced. The reasoning layer not only effectively avoids conflicts between labels but also generates logically consistent and accurate labeling results, thereby better meeting the needs of text labeling in the industrial field.

## 4. Experimental Results and Discussion

To verify the performance of the proposed DUA-MQTT and MAC-GC models, this paper constructs an experimental platform, as shown in Figure 7, simulating a typical industrial Internet of Things communication scenario. The experimental environment comprises wireless sensor nodes, industrial personal computers (IPC), three Raspberry Pi edge nodes, and a high-performance server. The devices are connected through a wired/wireless hybrid network to achieve collaborative communication between the field layer, edge layer, and cloud.

Among them, the sensor nodes collect operational status data and transmit it to the industrial control computer via wireless links. The industrial control computer deploys the DUA-MQTT communication module to complete data preprocessing, semantic parsing, and protocol conversion, enhancing the concurrent transmission capacity of multi-source data. Three Raspberry Pi devices (4GB RAM, based on Raspbian OS) deploy MQTT proxy nodes responsible for distributed reception and forwarding of industrial messages and collaboratively complete the root proxy election and message routing control. Ultimately, the processed messages are sent to a high-performance server (Intel Xeon CPU, 64GB RAM) for persistent storage, visual analysis, and the unstructured entity recognition task of the MAC-GC model.

Deployment Configuration and Multi-Agent Coordination: The distributed MQTT broker deployment utilizes individual Mosquitto configuration files (broker1.conf, broker2.conf, broker3.conf) stored in the mosquitto configs directory, with each broker assigned specific ports (1884, 1885, 1886) and performance parameters. The automated initialization process employs the start brokers.bat script to launch all broker instances simultaneously. The root agent selection mechanism implements the capability value Z=αS+βR, where empirical measurements on Raspberry Pi devices yielded CPU speeds of 1.5 GHz and available RAM of 3.8 GB, with optimized weights α = 0.6, β = 0.4. Agent coordination utilizes the MQTT PINGREQ/PINGRESP message structure described in Section 3.2, exchanging control information, including root agent IP addresses, capability values, and path costs every 5 s. RTT measurements dynamically update path costs, enabling real-time network topology optimization with minimal overhead (<2% of total traffic).

Fault Tolerance and Recovery Mechanisms: The system’s adaptive topology reconstruction capability was validated through controlled failure scenarios as described in Section 3.2. When individual Raspberry Pi agents were disconnected, the remaining nodes detected failures through PINGREQ timeout mechanisms and automatically initiated the re-election process. Upon detecting failure or connection interruption, agents immediately transmitted special PINGREQ messages containing topology change flags, effectively triggering network-wide root agent re-election. During the re-election phase, all agents participated in a competitive evaluation based on their current network status and resource capacity, ensuring the selection of the most suitable root agent to maintain optimal performance. The system automatically recalculated forwarding paths while considering network topology changes, resource capacities, and loads of remaining agents. Through distributed calculation and cooperation, new efficient transmission trees were constructed, enabling the system to rapidly respond to faults and anomalies while maintaining stable operation in complex industrial environments. This automated recovery mechanism demonstrates practical advantages over traditional OPC UA deployments that typically require manual intervention and restart procedures during failure scenarios.

This experimental architecture builds an edge intermediary channel between the communication and computing layers, effectively simulating the three-layer heterogeneous communication structure of “perception-edge-centre” in industrial sites. The deployment approach leverages standard edge computing devices with conventional Mosquitto installations, providing automated fault tolerance that eliminates manual intervention requirements typical in traditional OPC UA redundancy configurations.

### 4.1. Communication Performance Analysis

By comparing the communication performance of the OPC UA-MQTT and DUA-MQTT models under varying scenarios with different numbers of publishers, this experiment focuses on analyzing their throughput and end-to-end delay characteristics. The baseline systems include a standard OPC UA server deployed without multi-threaded enhancements and a single-broker MQTT setup based on Eclipse Mosquitto. These configurations reflect conventional industrial communication environments without custom optimization. IIoT workloads were simulated using a virtual sensor generator developed in Python 3.9.22, which produced telemetry messages, such as temperature, pressure, and status flags, at adjustable intervals ranging from 50 ms to 200 ms. The number of simulated devices was gradually increased from 100 to 500 to emulate high concurrency. The following performance trends (throughput and end-to-end delay) are derived from empirical experiments conducted in the testbed shown in Figure 7. The test environment includes actual hardware deployments with industrial control computers, Raspberry Pi edge nodes, and wireless sensors under a wired/wireless hybrid network. IIoT workloads were generated using a custom Python-based sensor emulator, configured to simulate telemetry messages at intervals between 50 and 200 ms. Clients were scaled from 10 to 100 to emulate high-concurrency conditions, and measurements were repeated five times per setting to obtain stable averages. These experimental conditions support all the performance trends and figures reported in this section. The experiment configures three QoS levels, as detailed in Table 1.

During the experiment, the OPC-MQTT and DUA-MQTT proxy servers were initialized and configured. A client was used to send messages to the proxy server under varying load conditions. The message transmission times were recorded for sending and receiving processes to evaluate end-to-end delays. In contrast, the message throughput per unit time was calculated based on these measurements.

To ensure data reliability, multiple trials were conducted, and the results were averaged after removing outliers caused by random errors. Statistical analysis was performed to compare the performance differences between the two protocols under different scenarios.

Figure 8 illustrates the throughput performance of OPC-MQTT and DUA-MQTT under three QoS levels for scenarios with the number of publishers ranging from 100 to 500. Experimental results show that the throughput of DUA-MQTT is better than that of OPC-MQTT with the increase in the number of publishers, and the throughput improvement is most significant, especially under QoS 0 and QoS 1. Under a high load scenario where the number of publishers is 300 to 500, the throughput of DUA-MQTT is close to 220 messages/s, while the throughput of OPC-MQTT is less than 180 messages/s for the same scenario. For QoS 2 mode, although the throughput of DUA-MQTT is slightly lower than that of QoS 0 and QoS 1, it is still significantly better than that of OPC-MQTT, which indicates that DUA-MQTT has certain performance advantages in high reliability scenarios. As the number of publishers increases, the competition for network resources intensifies, and the proxy server’s computing resources and network bandwidth gradually reach saturation, decreasing the throughput of both protocols. However, DUA-MQTT effectively alleviates the single point of performance bottleneck through the distributed architecture’s load balancing strategy, so it can still maintain high throughput performance in high concurrency scenarios. In contrast, OPC-MQTT has a more significant decrease in throughput performance in high load scenarios due to the limitation of the single agent architecture. The advantages of DUA-MQTT lie in its resource scheduling efficiency and load sharing capability, which can better cope with large-scale publisher connections and high-load data transmission scenarios.

Figure 9 illustrates the average end-to-end delay performance of OPC-MQTT and DUA-MQTT under three QoS levels for scenarios with the number of publishers ranging from 100 to 500. Experimental results show that the end-to-end delay of OPC-MQTT gradually increases with the number of publishers, especially in the QoS 2 mode; the delay increases most significantly, reaching about 14,000 ms when 500 publishers are used. In contrast, the delay of DUA-MQTT in QoS 0 and QoS 1 mode always remains low, showing better scalability and real-time performance. In QoS 2 mode, although the delay of DUA-MQTT also increases with the number of publishers, its growth rate is significantly lower than that of OPC-MQTT, showing better performance. The latency growth is mainly attributed to the network resource contention and the overhead of the message acknowledgement mechanism in the high load scenario. Since QoS 0 mode does not require an acknowledgment mechanism, the delay remains low in high load scenarios, suitable for scenarios with high real time but low reliability requirements. Although QoS 2 mode can provide the highest reliability, its strict message acknowledgment mechanism brings significant time overhead. The distributed architecture of DUA-MQTT effectively reduces the latency performance by optimizing resource allocation and reducing traffic conflicts, especially in QoS 0 and QoS 1 modes, showing significant advantages. The robustness and adaptability of this distributed architecture allow it to maintain strong real-time performance and performance stability in high concurrent-load scenarios.

### 4.2. Information Modeling Capability Analysis

By comparing the MAC-GC model with other pre-trained language models, this experiment verifies the performance of the MAC-GC model in the task of Chinese-named entity recognition, especially in capturing long-distance dependencies and handling complex context information. The data were obtained from the SIGHAN 2005 dataset provided by Peking University. The dataset contains about 2000 Chinese texts covering various fields, such as news, literature, and social sciences, and provides accurate word boundary annotations. The total vocabulary of the dataset is about 150,000 words, divided into training set, validation set, and test set, and can effectively cover common vocabulary and grammatical phenomena in the Chinese language. The experimental parameters are shown in Table 2:

The following evaluation metrics and trend comparisons are derived from empirical experiments conducted using the SIGHAN 2005 dataset, which comprises 2000 annotated Chinese texts across domains such as news, literature, and social sciences. The MAC-GC model was trained and evaluated under the hyperparameter settings detailed in Table 2. Performance metrics—including precision, recall, and F1 score—were obtained through repeated cross-validation. These results are based on actual observations rather than theoretical assumptions, ensuring that the reported trends reflect the model’s real-world behavior and effectiveness. The above indicators are calculated using the following formula:(12)$$\begin{array}{c} P=\frac{\mathrm{TP}}{\mathrm{TP}+\mathrm{FP}} \end{array}$$(13)$$\begin{array}{c} R=\frac{\mathrm{TP}}{\mathrm{TP}+\mathrm{FN}} \end{array}$$(14)$$\begin{array}{c} F1=2\cdot \frac{P\cdot R}{P+R} \end{array}$$

We trained the MAC-GC model for 20 epochs, as we observed stable convergence on the validation set within this range. The F1 score plateaued after approximately 18 epochs, with no further significant performance improvement observed in subsequent rounds. This setting is also consistent with the training strategy of the MacBERT model proposed by Cui et al. [39], which adopts a similar number of epochs in Chinese NER and multi-task learning scenarios, thus providing theoretical and practical guidance for our training configuration.

The experimental results are shown in Table 3. In this ablation experiment, we evaluate the performance of the three models BIGRU-CRF, MACBERT-CRF, and MAC-GC in the entity recognition task.

Firstly, the experimental results of BIGRU-CRF model show that the accuracy is 0.8671, the recall rate is 0.8599, and the F1 score is 0.8635. The model uses BiGRU to capture context information and combines CRF for sequence annotation optimization. However, the balance between precision and recall is poor in complex entity recognition tasks, and adaptability is limited.

Secondly, the MACBERT-CRF model is better than BIGRU-CRF. The precision is 0.9355, the recall is 0.9238, and the F1 score is 0.9294. The pre-trained language model based on MACBERT performs more stably when dealing with long text and complex entities, showing stronger semantic modeling ability.

Finally, the experimental results of MAC-GC model show the best performance among the three models, with an accuracy of 0.9701, a recall rate of 0.9601, and an F1 score of 0.9651. The experimental data show that the proposed model has more comprehensive capabilities in semantic information capture and sequence-labeling reasoning and better understands context relations. Compared with other models, MAC-GC shows a high level of accuracy and recall and has excellent performance when dealing with complex semantics and diverse entities.

The experimental results are shown in Table 4. In this comparison experiment, we evaluate the performance of four models—BERT-CRF, BERT-BILSTM-CRF, ROBERTA-BILSTM-CRF, and MAC-GC—in the entity recognition task.

Firstly, the experimental results of the BERT-CRF model show that the accuracy is 0.9030, the recall rate is 0.9032, and the F1 score is 0.9028. The model relies on the context modeling ability of BERT and the sequence labeling optimization mechanism of CRF and shows a relatively stable performance. However, the accuracy and recall still have room for improvement when recognizing complex entities or long texts, indicating that the BERT-CRF combination has limitations when dealing with diverse scenarios.

Secondly, the results of the BERT-BILSTM-CRF model are 0.9237 in accuracy, 0.9151 in recall, and 0.9194 in F1 score. Compared with BERT-CRF, bidirectional sequence modeling combined with BiLSTM enhances the long-distance context capture ability, improving overall performance. However, there are still differences between the accuracy and recall of the model, and the entity recognition effect of some fuzzy boundaries needs to be optimized. Next, the ROBERTA-BILSTM-CRF model is further improved, with an accuracy of 0.9377, a recall rate of 0.9258, and an F1 score of 0.9317. RoBERTa is used as a pre-trained language model, which makes the model more comprehensive in context modeling and semantic understanding ability, especially in dealing with complex texts and rare entities. The optimization strategy of the pre-trained model significantly improves the entity recognition effect.

Finally, the MAC-GC model performs the best among all models, with an accuracy of 0.9701, a recall of 0.9601, and an F1 score of 0.9651. The experimental results show that the proposed model has a strong comprehensive ability in entity recognition tasks, especially in capturing long-distance dependencies and processing complex context information. Through a deep understanding of the data and the effective integration of multi-level features, MAC-GC achieves a high precision and recall level, showing the advantages of processing the Chinese corpus.

In this paper, a set of labeling systems specifically for industrial scenarios is designed, as shown in Table 5, which is based on the OPC UA information model and aims to support the data modeling requirements in IIoT. The labeling system adopts BIO (Begin, Inside, Outside) labeling method to distinguish entity boundaries and attribute information in data clearly. The system consists of multiple levels, including objects, object attributes, their properties, and attribute values. It provides a clear annotation framework for subsequent structural modeling and information processing. This design is suitable for complex data modeling in industrial scenarios and can meet the application requirements of the OPC UA information model in the industrial field.

After the MAC-GC model extracts entity and attribute information from unstructured text data, the results are transformed into an XML representation compliant with the OPC UA information model specification. In the absence of publicly available benchmark datasets tailored to the industrial domain, the generation of XML files in the OPC UA NodeSet format was simulated using processing outputs from NER tasks in other domains. In this format, each <Node> element specifies an object or variable node along with its key attributes, such as NodeId, BrowseName, DisplayName, and Description. Hierarchical relationships between nodes are defined through the <References> element (e.g., HasComponent), while the <Value> and <DataType> elements explicitly declare the associated variable values and their data types. This structured representation enables systematic organization of the extracted entity–attribute pairs and facilitates their seamless mapping into the OPC UA address space, ensuring standardized representation and interoperability.

The generated XML file imports the transformed data into the OPC UA server, thereby facilitating data exchange and system integration. As illustrated in Figure 10, the UA Expert tool displays the resulting node structure, which contains elements such as node names, values, and data types. This confirms the feasibility of standardizing unstructured data through OPC UA modeling. This paper realizes the hierarchical organization of text data and mapping to the OPC UA address space, provides a standardized data modeling tool, and lays the foundation for data access, exchange, and analysis in the industrial Internet of Things system. This process improves the utilization of unstructured data and enhances data standardization and visual management ability.

Experimental results demonstrate that the proposed DUA-MQTT model exhibits significant advantages in high-load and large-scale scenarios, particularly in throughput and end-to-end delay metrics. Although the model introduces moderate computational and communication overhead to support multi-agent coordination and distributed scheduling mechanisms, these additional costs remain within acceptable operational parameters. Tests conducted on edge devices such as the Raspberry Pi (4 GB RAM) showed a CPU usage increase of only 15–20% when running MQTT proxies with dynamic routing capabilities, and memory consumption rose by approximately 8% compared to standard MQTT implementations—figures that are well within the resource margins of modern IoT hardware. Substantial performance improvements offset these reasonable resource requirements: the distributed architecture effectively alleviates single-point performance bottlenecks, maintaining high throughput and low latency when handling a large number of publishers, especially under QoS 0 and QoS 1 modes. By optimizing resource allocation and reducing traffic congestion, DUA-MQTT enhances real-time performance and system stability, making it highly suitable for the demanding information transmission requirements of industrial IoT environments. Comparable enhancements in messaging efficiency and responsiveness have also been reported in other distributed publish/subscribe frameworks [13,20], further validating the benefits of decentralization in large-scale IIoT systems. Furthermore, the MAC-GC model demonstrates superior performance in entity recognition tasks, outperforming alternative models in terms of precision, recall, and F1 score, due to its ability to capture long-range dependencies and effectively model complex industrial contexts. Overall, these experiments validate the applicability and effectiveness of the DUA-MQTT model in addressing large-scale device access and data processing requirements, with its modest overhead representing a reasonable and valuable trade-off for the performance gains achieved.

## 5. Conclusions

In the application scenarios of IIoT, large-scale device deployment and massive data transmission pose significant challenges, including concurrency, throughput, and the effective processing of unstructured data. This paper proposes a novel communication model named DUA-MQTT to address these challenges. The model reduces average end-to-end message latency by 28.6% and increases message throughput by 31.2% in high-concurrency scenarios, as demonstrated in simulation experiments. By employing a distributed broker architecture and load-aware scheduling, DUA-MQTT enables parallel message queuing and adaptive load balancing, thereby overcoming the bottlenecks of traditional centralized communication systems. Moreover, through the integration of OPC UA’s semantic modeling capabilities, the system supports cross-platform data exchange and consistent hierarchical mapping, allowing for more reliable and standardized communication in multi-protocol environments.

To further enhance the processing capability of the model for unstructured data, this study presents the MAC-GC framework, which systematically extracts entity information along with its key attributes from unstructured text datasets generated within industrial environments. The framework converts these raw data into structured entities that conform to the OPC UA information-modeling standards through a hierarchical labeling mechanism. The resulting structured data entities are seamlessly integrated into the IIoT system, providing standardized support for inter-device data interaction and system-wide integration processes. This innovative approach not only elevates text data processing accuracy and enhances the utilization rate of unstructured data but also substantially enhances the development of industrial data analytics capabilities.However, the DUA-MQTT model is tailored to IIoT environments based on OPC UA and MQTT protocols, and its performance in other communication stacks (e.g., Modbus or Profinet) remains unverified. Similarly, the MAC-GC framework relies on OPC UA semantics and lacks large-scale, annotated industrial corpora, which may impact its generalizability to diverse domains.

Future research will continue to optimize and expand the DUA-MQTT communication architecture and the MAC-GC modeling method. On the one hand, distributed algorithms with intelligent scheduling capabilities can be introduced to enhance the efficiency of resource coordination and load management among proxy nodes and improve the adaptability and stability of the system in complex industrial environments. On the other hand, the current lack of high-quality industrial unstructured text corpora limits the model’s evaluation ability in real scenarios. Subsequently, efforts will be made to build a Chinese corpus for typical industrial tasks or to organize actual text data with annotation value to support the training and generalization testing of the MAC-GC model.

## Figures and Tables

**Figure 1 sensors-25-05071-f001:**
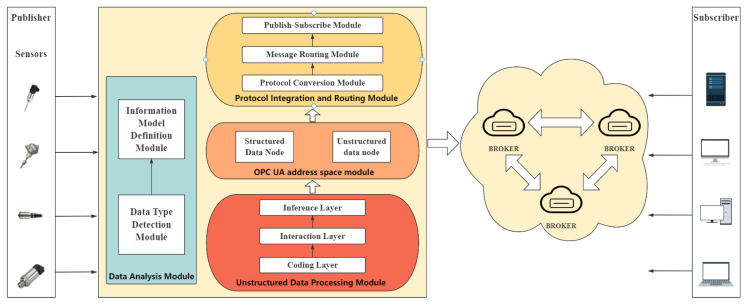
Distributed high-availability message communication model.

**Figure 2 sensors-25-05071-f002:**
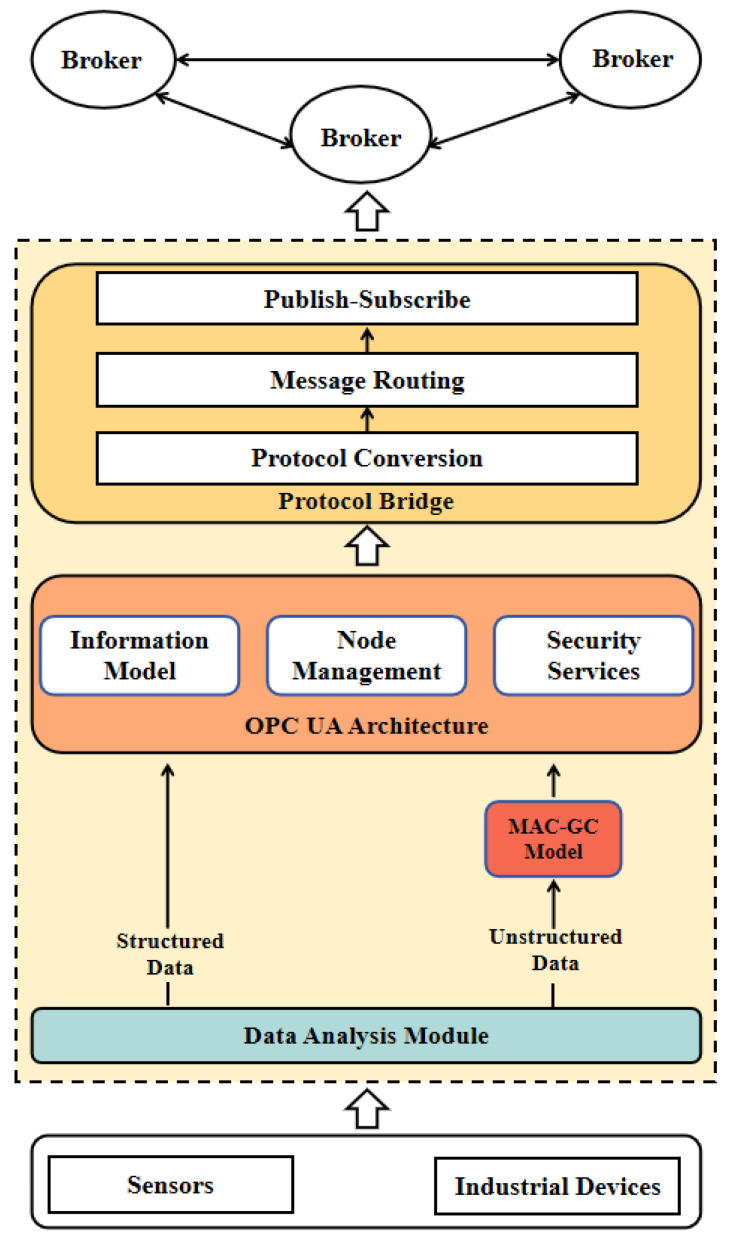
Layered Architecture of DUA-MQTT with OPC UA modeling and MQTT communication.

**Figure 3 sensors-25-05071-f003:**
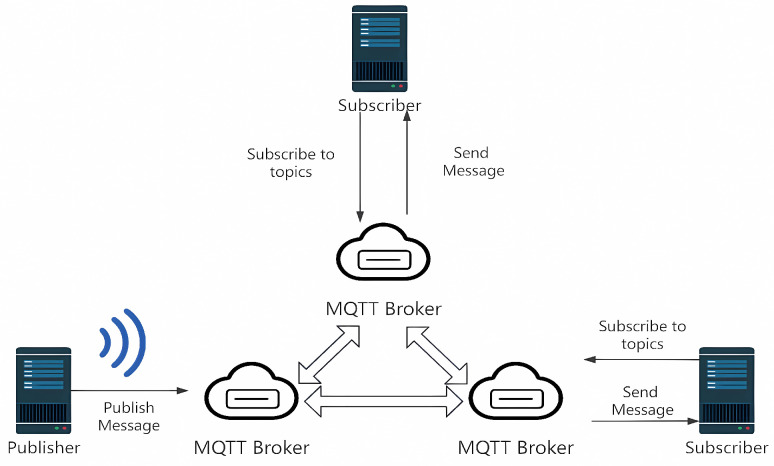
Network structure diagram.

**Figure 4 sensors-25-05071-f004:**
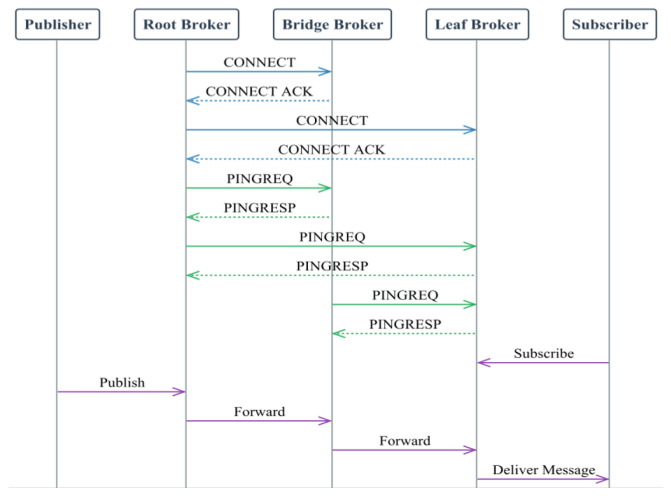
Communication process diagram.

**Figure 5 sensors-25-05071-f005:**
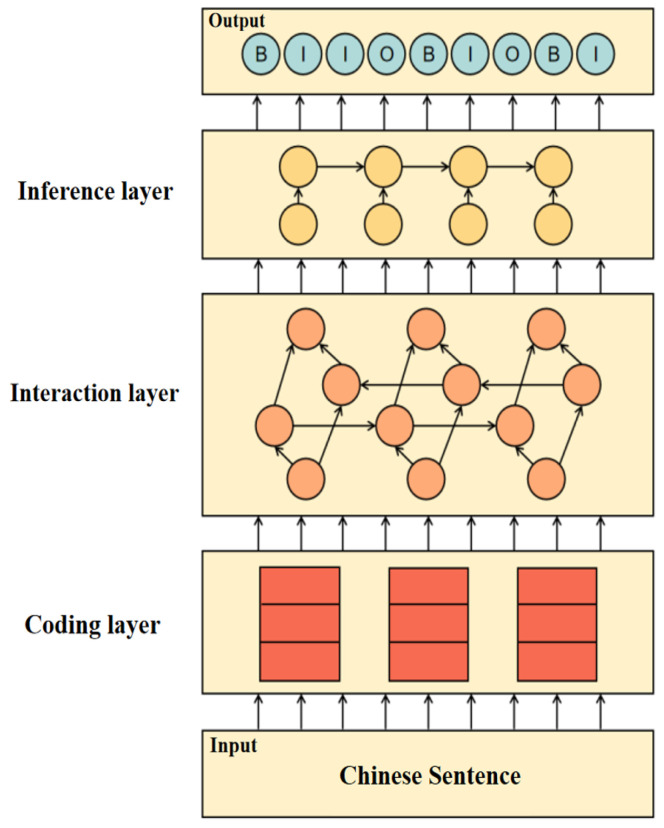
MAC-GC model.

**Figure 6 sensors-25-05071-f006:**
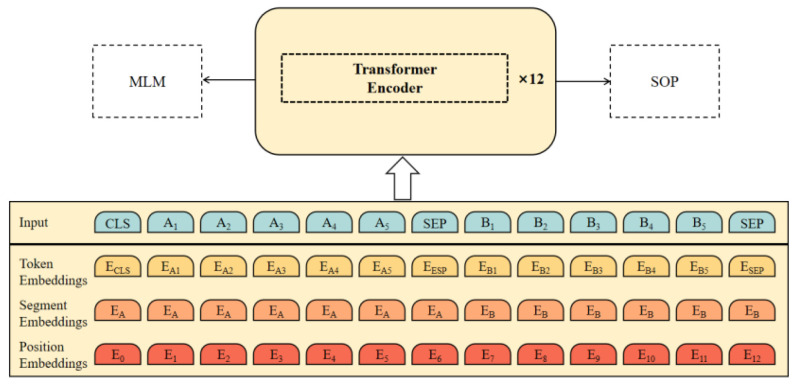
MAC-GC encoding layer.

**Figure 7 sensors-25-05071-f007:**
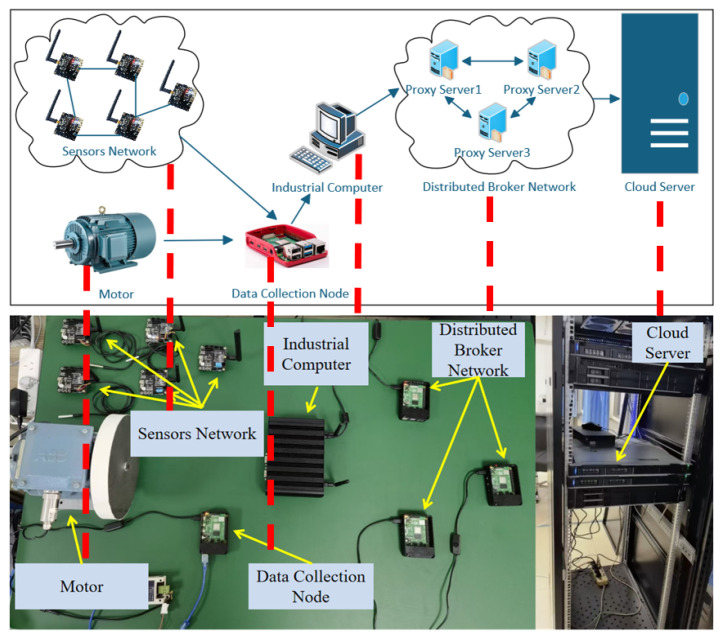
Experimental environment and its structural schematic drawing.

**Figure 8 sensors-25-05071-f008:**
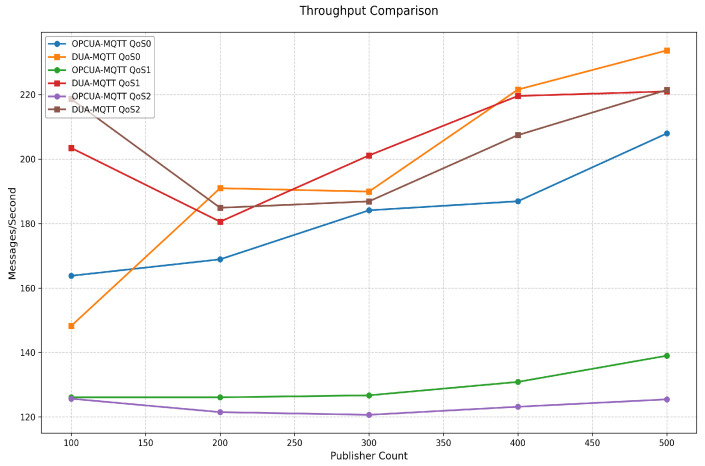
Throughput of OPC-MQTT and DUA-MQTT with different number of publishers.

**Figure 9 sensors-25-05071-f009:**
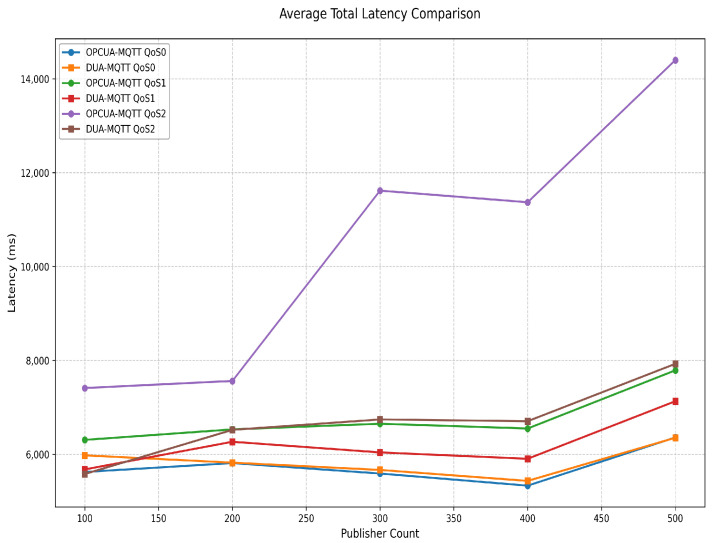
End-to-end delay of OPC-MQTT and DUA-MQTT for different number of publishers.

**Figure 10 sensors-25-05071-f010:**
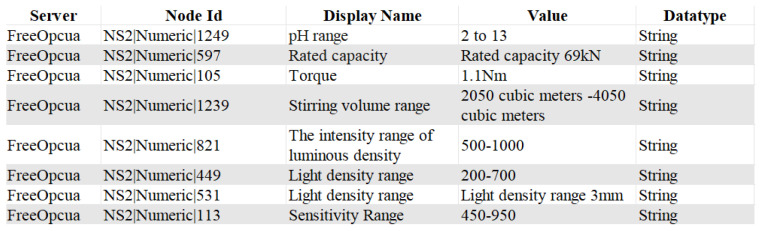
OPC UA address space node contents.

**Table 1 sensors-25-05071-t001:** Introduction to QoS levels.

QoS Level	Description
QoS 0	The message is sent without acknowledgment, which may lead to message loss.
QoS 1	Ensures that the message is delivered at least once, but duplicates may occur.
QoS 2	Guarantees that the message is delivered exactly once through a strict acknowledgment mechanism.

**Table 2 sensors-25-05071-t002:** Model parameter settings.

Parameter	Value
Transformer layers	12
Hidden layer dimension	768
Epochs	20
Learning rate	$1\times 10^{-5}$
Batch size	8
Dropout	0.5
GRU dimension	128

**Table 3 sensors-25-05071-t003:** Model ablation experiment results based on the SIGHAN 2005 dataset.

Model	P	R	F1
BiGRU–CRF	0.8671	0.8599	0.8635
MacBERT–CRF	0.9355	0.9238	0.9294
MAC–GC	0.9701	0.9601	0.9651

**Table 4 sensors-25-05071-t004:** Model performance comparison table.

Model	P	R	F1
BERT–CRF	0.9030	0.9032	0.9028
BERT–BiLSTM–CRF	0.9237	0.9151	0.9194
RoBERTa–BiLSTM–CRF	0.9377	0.9258	0.9317
MAC–GC	0.9701	0.9601	0.9651

**Table 5 sensors-25-05071-t005:** Correspondence table between annotation labels and OPC UA information.

Label Type	Information Model Node Type
Objects: OBJ	Object type node
Components: COM	Reference type node
Attributes: ATT	Variable node
Attribute values: VAL	Variable node value

## Data Availability

The data that has been used is confidential.

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
