# Peer review of "DUA-MQTT: A Distributed High-Availability Message Communication Model for the Industrial Internet of Things"

_sensors, 2025, doi:10.3390/s25165071_

Round 1
Reviewer 1 Report
Comments and Suggestions for Authors
This paper proposes a hybrid communication and modeling solution for the Industrial Internet of Things, but it requires major revisions to improve clarity, technical depth, and reproducibility. Here is the list of my comments:
- the paper highlights challenges related to concurrency and throughput in IIoT communication systems, the problem formulation remains vague. Please clearly define the research problem and articulate the real-world implications and urgency of addressing it. Include quantifiable limitations of existing OPC UA or MQTT-based systems that justify the need for DUA-MQTT.
- The rationale behind combining OPC UA and MQTT in the proposed DUA-MQTT model needs further elaboration. What are the theoretical underpinnings or architectural principles that support this hybrid approach? I read through the context of this paper but feel this problem is not well explained. Please include comparative discussion of alternative architectures and why DUA-MQTT is preferable.
- The MAC-GC component is described briefly, but its methodology is unclear. Please provide a detailed explanation of how hierarchical annotation is applied, how unstructured text is processed, and what techniques (e.g., NLP models, rule-based systems) are used to generate structured nodes. The authors can also refer to some related literature reviews such as “When Software Security Meets Large Language Models: A Survey” by Zhu et al. as a reference for proper techniques.
- The paper presents strong performance claims (e.g., 28.6% latency reduction), but lacks sufficient detail about the experimental setup. Please clarify what baseline systems were used for comparison and how were the IIoT workloads generated?
- There is a minor issue: some sections of the paper use general or ambiguous language (e.g., "improving utilization efficiency"). This paper can benefit from increased technical precision. Please define all terms and avoid vague description.
- This paper does not have sufficient discussion on whether the proposed method is secure or not. I would like to suggest the authors refer to some related work like “Network Traffic Fingerprinting for IIoT Device Identification: A Survey” by Sheng et al. and “Detecting vulnerability on IoT device firmware: A survey” by Feng et al. as reference for more comprehensive discussion.
- This paper lacks a critical reflection on its limitations. Please include a more detailed discussion on the limitations of the current model (e.g., constrained to specific protocols or domains).
Author Response
Dear Reviewer,
Thank you very much for your valuable and constructive comments on our manuscript. We sincerely appreciate the time and effort you invested in reviewing our work.
In response to your suggestions, we have carefully revised the manuscript. All recommended changes have been implemented, and each modification is documented in detail. For your convenience, a point-by-point response to your comments, along with the corresponding revisions, has been included in the appendix of this document.
We hope that the revised version meets your expectations and fully addresses your concerns. Please feel free to let us know if any additional improvements are needed.
Thank you once again for your thoughtful feedback and kind support.
Sincerely,
Mengjia Lian (Corresponding Author)
Longyan University, China
Email: AIoTeam@126.com

Reviewer 2 Report
Comments and Suggestions for Authors
My revision is in the attached file

Reviewer 3 Report
Comments and Suggestions for Authors
While the study presents a quite detailed overview of the proposed DUA-MQTT message communication model for the industrial Internet of Things it could benefit from the following suggestions in terms of clarity and scientific soundness:
- Please consider clarifying all Figures by magnifying the text to offer an improved aspect to the reader. Regarding Figure 1, you could include the word “Sensors” in the left column, magnify the structure part and the included text while shrinking the “broker” part.
- The Algorithm 1 “DUA-MQTT Network Construction and Message Forwarding” table could be moved to an appendix at the end of the article.
- On what grounds do you suggest 20 training rounds? You may add more discussion and even scientific literature/references to back up your suggestion.
- Please consider rephrasing the title of Table 4 to “Model performance comparison table” just to be more specific.
- Instead of the word “(ours)” you may use “(suggested or under estimation…)”
- You may add more discussion and references within the text to support your implication in line 667.
- What are the overheads (such as computing software, data storage, computing efficiency, etc.) of the proposed scheme? More discussion could be added.
Author Response

(The authors gave the same response as above.)

Round 2
Reviewer 1 Report
Comments and Suggestions for Authors
The authors have addressed all my concerns before. A minor issue: Figure 10 is a screen shot, which should be replaced by a figure. We should not use screen shots in a paper.
Author Response
Thank you for your valuable comments and suggestions on our manuscript. We have carefully revised the paper in accordance with your feedback and have provided a point-by-point response to each of your comments. Please find the detailed responses in the attached document.

Reviewer 2 Report
Comments and Suggestions for Authors
The revision adequately reflects the comments. The reviewer's only comment is on the quality of the Figures; they need to be enhanced for better readability.
Author Response
We have revised our manuscript according to the reviewer’s comments and provided detailed point-by-point responses. Please find our responses and the revised manuscript attached for your kind consideration.
